# Multi-Echo Complex Quantitative Susceptibility Mapping and Quantitative Blood Oxygen Level-Dependent Magnitude (mcQSM + qBOLD or mcQQ) for Oxygen Extraction Fraction (OEF) Mapping

**DOI:** 10.3390/bioengineering11020131

**Published:** 2024-01-29

**Authors:** Junghun Cho, Jinwei Zhang, Pascal Spincemaille, Hang Zhang, Thanh D. Nguyen, Shun Zhang, Ajay Gupta, Yi Wang

**Affiliations:** 1Department of Biomedical Engineering, State University of New York at Buffalo, Buffalo, NY 14228, USA; 2Department of Radiology, Weill Cornell Medicine, New York, NY 10065, USApas2018@med.cornell.edu (P.S.);

**Keywords:** oxygen extraction faction, quantitative susceptibility mapping, quantitative blood oxygen level-dependent imaging, multi-echo complex quantitative susceptibility mapping and quantitative blood oxygen level-dependent magnitude, QSM + qBOLD, QQ, mcQQ, deep learning, magnetic resonance imaging, MRI

## Abstract

Oxygen extraction fraction (OEF), the fraction of oxygen that tissue extracts from blood, is an essential biomarker used to directly assess tissue viability and function in neurologic disorders. In ischemic stroke, for example, increased OEF can indicate the presence of penumbra—tissue with low perfusion yet intact cellular integrity—making it a primary therapeutic target. However, practical OEF mapping methods are not currently available in clinical settings, owing to the impractical data acquisitions in positron emission tomography (PET) and the limitations of existing MRI techniques. Recently, a novel MRI-based OEF mapping technique, termed QQ, was proposed. It shows high potential for clinical use by utilizing a routine sequence and removing the need for impractical multiple gas inhalations. However, QQ relies on the assumptions of Gaussian noise in susceptibility and multi-echo gradient echo (mGRE) magnitude signals for OEF estimation. This assumption is unreliable in low signal-to-noise ratio (SNR) regions like disease-related lesions, risking inaccurate OEF estimation and potentially impacting clinical decisions. Addressing this, our study presents a novel multi-echo complex QQ (mcQQ) that models realistic Gaussian noise in mGRE complex signals. We implemented mcQQ using a deep learning framework (mcQQ-NET) and compared it with the existing QQ-NET in simulations, ischemic stroke patients, and healthy subjects, using identical training and testing datasets and schemes. In simulations, mcQQ-NET provided more accurate OEF than QQ-NET. In the subacute stroke patients, mcQQ-NET showed a lower average OEF ratio in lesions relative to unaffected contralateral normal tissue than QQ-NET. In the healthy subjects, mcQQ-NET provided uniform OEF maps, similar to QQ-NET, but without unrealistically high OEF outliers in areas of low SNR, such as SNR ≤ 15 (dB). Therefore, mcQQ-NET improves OEF accuracy by more accurately reflecting realistic Gaussian noise in complex mGRE signals. Its enhanced sensitivity to OEF abnormalities, based on more realistic biophysics modeling, suggests that mcQQ-NET has potential for investigating tissue variability in neurologic disorders.

## 1. Introduction

Mapping metabolic oxygen consumption is essential for assessing brain tissue viability and function in cerebrovascular and neurodegenerative disorders, including Alzheimer’s disease (AD) [1], Parkison’s disease [2], dementia [3], and multiple sclerosis [4,5]. For instance, in stroke, a major therapeutic target is to salvage the penumbra—low perfusion yet viable tissue—using endovascular treatments [6,7]. Therefore, the precise identification of the penumbra is essential for clinical decision making. Elevated oxygen extraction fraction (OEF) directly indicates the penumbra’s presence, whereas current perfusion/diffusion mismatch approaches cannot optimally identify the penumbra [8,9]. OEF also serves as a promising biomarker for investigating AD, including identifying cognitive impairment through decreased OEF [10] and assessing vascular risks through elevated OEF [11,12]. Despite these clinical needs, there is currently no OEF mapping technique available in clinical settings. Positron emission tomography (PET) using three ^15^O tracers is the current standard [13]. However, it is not commonly used clinically due to the complexity of required radioligand injections and the necessity for on-site production of ^15^O tracers because of their short 2 min half-life [13].

To address the limitations of PET, widely available MRI has been investigated for quantitative mapping of OEF. MRI-based OEF methods have mainly focused on the strong paramagnetic effects of deoxyhemoglobin in blood on the magnitude signal, including calibrated functional MRI (cfMRI) [14,15], T2-based methods [16,17], and quantitative blood oxygen level-dependent magnitude (qBOLD) [18,19,20]. However, these methods have their own issues. First, cfMRI estimates OEF from its empirical relationship with R2*, but it suffers from poor sensitivity and requires impractical vascular challenges, including hypercapnia and hyperoxia [21]. Second, T2-based methods estimate OEF from venous blood T2 mapping using velocity-selective spin labeling [16], but they struggle with the problematic venous blood signal separation [17]. Third, qBOLD estimates OEF from magnitude signal decay, assuming deoxyhemoglobin is distributed in cylinders, but its poorly conditioned model inversion leads to high uncertainty in OEF estimation [22]. Additionally, phase signal modeling has been also introduced for OEF mapping, including quantitative susceptibility mapping (QSM) [21,23,24]. The QSM-based methods separate the voxel-wise magnetic susceptibility into contributions of deoxyhemoglobin in blood (OEF effect) and its surrounding non-blood neural tissue source.

Recently, a novel MRI-based OEF mapping technique that combines QSM and qBOLD (QSM + qBOLD = QQ) has been introduced [25]. QQ accounts for the OEF effect on both the phase (by QSM) and magnitude (by qBOLD) of a routinely accessible multi-echo gradient echo (mGRE) dataset [25]. It also eliminates unrealistic assumptions in each QSM (e.g., linear relationship between blood flow and blood volume) and qBOLD (e.g., constant non-blood neural tissue susceptibility throughout the whole brain). The robustness of QQ was subsequently improved by using data processing algorithms, including an unsupervised clustering algorithms [26]. By eliminating impractical vascular challenges and utilizing a single routine MRI sequence, QQ has demonstrated its clinical feasibility in neurologic disorders, such as ischemic stroke [27]. Moreover, a deep learning approach further improved the robustness against noise and the reconstruction speed in QQ [28].

Current QQ data processing approaches, which encompass both machine learning and deep learning, commonly assume Gaussian noise for both QSM and mGRE magnitude signals [26,28]. However, this assumption may not always be valid, especially in low signal-to-noise ratio (SNR) regions, such as stroke lesions. This is because mGRE magnitude signals exhibit Rician noise, which may not approximate Gaussian noise in low SNR scenarios [29], and QSM may not be governed by Gaussian noise owing to multiple nonlinear steps required for QSM estimation from mGRE phase signals with non-Gaussian noise [30]. Such problematic assumptions can compromise OEF accuracy, particularly in disease-related abnormal lesions, which possibly influence clinical decisions for neurologic disorders.

This study introduces a novel multi-echo complex QQ approach (mcQQ) that accurately assumes Gaussian noise in the mGRE complex signals. Given that measurement noise substantially affects OEF accuracy in QQ approaches [26], a more realistic consideration of noise in mcQQ is expected to enhance OEF accuracy. For a fair comparison with the current deep learning-based QQ (QQ-NET) [28], we implemented a deep learning approach to solve mcQQ (mcQQ-NET) using the same training and test schemes as well as the same datasets as QQ-NET. We then compared the proposed mcQQ-NET with QQ-NET in simulations, ischemic stroke patients, and healthy subjects.

## 2. Materials and Methods

### 2.1. Data Acquisition

The datasets utilized in this study were retrospectively acquired from a previous QQ-NET study [28]. The study was approved by the local Institutional Review Board and involved MRI scans of 34 ischemic stroke patients (occurring within a unilateral cerebral artery territory) between 6 h and 42 days post stroke. The scans were conducted on a clinical 3T scanner (GE MR Discovery 750) utilizing a 32-channel brain receiver coil. The 3D mGRE imaging protocol was applied, with the following parameters: 0.47 × 0.47 × 2.0 mm^3^ voxel size, eight equally spaced echoes (TE_1_/ΔTE/TE_8_ = 4.5/5/39.5 ms), TR= 42.8 ms, bandwidth = 244.1 Hz/pixel, 20^o^ flip angle, and 5 min 15 s scan time. Further, DWI (24 cm FOV, 0.94 × 0.94 × 3.2 mm^3^ voxel size, 1953.1 Hz/pixel bandwidth, 0, 1000 s/mm^2^ b-values, TE = 71 ms, TR = 3000 ms, and four signal averages) and a T1-weighted fluid-attenuated inversion recovery sequence (24 cm FOV, 0.5 × 0.5 × 5 mm^3^ voxel size, TE = 23.4 ms, TR = 1750 ms) were used.

To check whether a network trained on simulated stroke datasets could reliably produce uniform OEF maps in healthy subjects without generating false negatives, MRI scans of four healthy subjects (age 31 ± 6 years) were also retrospectively obtained from the QQ-NET study [28]. These subjects underwent MRI scans on a 3T GE scanner, using 3D mGRE with imaging parameters that matched those used for the stroke patients.

### 2.2. Data Processing: QSM

QSM reconstruction consisted of the following steps [30]: calculating the total field ($f_{T}$) with a nonlinear fit of the mGRE, estimating the local field from the background field ($f_{B}$) via the projection onto dipole fields (PDFs) method, and obtaining susceptibility through morphology-enabled dipole inversion with automatic uniform CSF zero-reference algorithm. The FSL FLIRT algorithm was used to co-register all images with the QSM maps [31,32]. QQ-NET used the QSM maps as the network’s inputs for OEF estimation, whereas mcQQ-NET did not.

### 2.3. Data Processing: OEF Using Multi-Echo Complex QQ (mcQQ)

In the mcQQ model, a nonlinear formulation is used when integrating the QSM-based and qBOLD-based OEF mapping methods to obtain OEF = $1-Y/Y_{a}$, where Y and $Y_{a}$ (=0.98) [25] denote venous and arterial oxygenation, respectively. The mGRE complex signal at the *j*’th echo with the compensation of the initial phase and background field contribution on phase, ($S_{j}$), can be modeled as Equation (1):(1)$$S_{j}=F_{qBOLD}\left(S_{0},Y,v,R_{2},\chi_{n},t_{j}\right)e^{i\omega_{0}t_{j}d*F_{QSM}\left(Y,v,\chi_{n}\right)}$$

In the magnitude term of Equation (1), the qBOLD models the OEF effect [19]:(2)$$F_{qBOLD}\left(S_{0},R_{2},Y,v,\chi_{n},t_{j}\right)={S_{0}\cdot e}^{-R_{2}\cdot t_{j}}\cdot F_{BOLD}\left(Y,v,\chi_{n},t_{j}\right)\cdot G(t_{j})$$
where $S_{0}$ is signal intensity at $t_{j}=0$; $R_{2}$ is the transverse relaxation rate (the microscopic field effect); $F_{BOLD}$ is the mesocropic field effect on mGRE signal due to deoxygenated blood $F_{BOLD}^{\;}\left(Y,v,\chi_{n},t\right)=exp\left[-v\cdot f_{s}\left(\delta \omega \cdot t\right)\right]$ [19]; v is the venous blood volume; $f_{s}$ is the signal decay due to the blood vessel network [19]; and δω is the characteristic frequency by the susceptibility difference between deoxygenated blood and the surrounding tissue: $\delta \omega \left(Y,\chi_{n}\right)=\frac{1}{3}\cdot \gamma \cdot B_{0}\cdot \left[\psi_{Hb}\cdot \Delta \chi_{Hb}\cdot \left(1-Y\right)+\chi_{ba}-\chi_{n}\right]$, where γ = 267.51 rad⋅s^−1^T^−1^ is the gyromagnetic ratio; $B_{0}$ is the main magnetic field; $\psi_{Hb}$ is the hemoglobin volume fraction (0.0909 with Hct = 0.357) [28]; $\Delta \chi_{Hb}$ is the susceptibility difference between deoxy and oxyhemoglobin (12,522 ppb) [29]; $\chi_{ba}$ is the fully oxygenated blood susceptibility (−108.3 ppb with tissue hematocrit Hct = 0.357) [30]; $\chi_{n}$ is the non-blood neural tissue susceptibility on a voxel-wise basis; and $G(t_{j})$ is the macroscopic field effect on mGRE signal [33].

In the phase term of Equation (1), the QSM-based method ($F_{QSM}$) distinguishes the venous blood deoxyhemoglobin’s susceptibility contribution (OEF effect) from $\chi_{n}$ on a voxel-wise basis, where $\omega_{0}$ is the Larmor frequency, d is the dipole kernel, and * is the convolution operator.
(3)$$F_{QSM}\left(Y,v,\chi_{n}\right)=\left[\frac{\chi_{ba}}{\alpha }+\psi_{Hb}\cdot \Delta \chi_{Hb}\cdot \left(-Y+\frac{1-\left(1-\alpha \right)\cdot Y_{a}}{\alpha }\right)\right]\cdot v+\left(1-\frac{v}{\alpha }\right)\cdot \chi_{n}$$
where α is the ratio between v and the total blood volume (0.77) [34].

### 2.4. Deep Neural Network for mcQQ (mcQQ-NET)

In order to account for realistic measurement noise (i.e., Gaussian noise in complex mGRE signals), mcQQ-NET introduces two modifications compared to the current QQ deep learning model (QQ-NET) [28]: changes in network structure and model loss.

Regarding the network structure, mcQQ-NET employs a combination of two Unet-based sub-networks (Figure 1) [35,36]. Each sub-network processes either the mGRE magnitude or phase input, whereas QQ-NET uses a single Unet to handle mGRE magnitude and QSM. In detail, for each sub-network of mcQQ-NET, the original U-net was modified to: (1) use zero-padding to maintain a uniform convolution layer output size, and (2) set the number of inputs to 8 for each sub-network (comprising 8 echo magnitude and phase signals, respectively) and outputs to 5 for the magnitude sub-network ($S_{0},R_{2},Y,v,\chi_{n}$) and 3 for the phase sub-network ($Y,v,\chi_{n}$). The setting for the numbers of outputs and inputs are based on how magnitude and phase signals can be modeled as functions of these parameters, as described by Equations (2) and (3). Each sub-network consists of an encoding and decoding path with 18 convolutional layers with 3 × 3 × 3 kernel, 4 max pooling layers with 2 × 2 × 2 kernel, 4 deconvolution layers with 2 × 2 × 2 kernel, 4 feature concatenations, and 1 convolutional layer with 1 × 1 × 1 kernel. Subsequently, the three common outputs of the two sub-networks ($Y,v,\chi_{n}$) were combined by utilizing a linear combination operation (gray arrow in Figure 1). Lastly, element-wise Tanh operation was applied to set the upper (max) and lower (min) bounds for the model parameters (blue arrow in Figure 1), based on physiological expectations for *Y* (0~100%) and *v* (0.5~5.5%), and CCTV results for the other parameters, mirroring the approach in QQ-NET.

Regarding the model loss, mcQQ-NET has a weighed sum of three losses:(1)L1 difference between the normalized truth and the output of mcQQ ($E_{L1}$):(4)$$E_{L1}={\left\|Z_{T}-Z_{O}\right\|}_{1}$$


Here, $Z_{T}=[z_{S_{0},T}z_{R_{2},T},z_{Y,T},z_{v,T},z_{\chi_{n},T}]$ and $Z_{O}=[z_{S_{0},O}z_{R_{2},O},z_{Y,O},z_{v,O},z_{\chi_{n},O}]$, where Z indicates z-score normalization (e.g., $z_{S_{0}}=\frac{S_{0}-\mu_{S_{0}\;}}{\sigma_{S_{0}}}$, where $\mu_{S_{0}}$ and $\sigma_{S_{0}}$ are the average and standard deviation of $S_{0}$) and the subscript “T” and “O” indicate the truth and output of the network, respectively.

(2)L1 difference of Y spatial gradient (∇) to preserve edge ($E_{Grad}$):(5)$$E_{Grad}={\left\|\nabla z_{Y,T}-\nabla z_{Y,O}\right\|}_{1}$$


Please note that $E_{L1}$ and $E_{Grad}$ in mcQQ-NET are identical to those in QQ-NET [28].

(3)The model loss to consider physical model consistency is represented by $\left(E_{Model}\right)$. Notably, $E_{Model}$ is different between mcQQ-NET and QQ-NET:

In mcQQ-NET:(6)$$E_{Model}={\left\|F_{qBOLD}\left(S_{0,T},Y_{T},v_{T},R_{2,T},\chi_{n,T}\right)e^{i\omega_{0}t_{j}d*F_{QSM}\left(Y_{T},v_{T},\chi_{n,T}\right)}\;\;-F_{qBOLD}\left(S_{0,O},Y_{O},v_{O},R_{2,O},\chi_{n,O}\right)e^{i\omega_{0}t_{j}d*F_{QSM}\left(Y_{O},v_{O},\chi_{n,O}\right)}\right\|}_{1}$$

In QQ-NET:(7)$$E_{Model}={\left\|S_{qBOLD}\left(S_{0,T},R_{2,T},Y_{T},v_{t},\chi_{n,T}\right)-S_{qBOLD}\left(S_{0,O},R_{2,O},Y_{O},v_{O},\chi_{n,O}\right)\;\right\|}_{1}\;\;+{\left\|\chi_{QSM}\left(Y_{T},v_{T},\chi_{n,T}\right)-\chi_{QSM}\left(Y_{O},v_{O},\chi_{n,O}\right)\;\right\|}_{1}$$

The total loss (E) is set as $E=E_{L1}+0.1\cdot E_{Model}+0.1\cdot E_{Grad}$.

To ensure a fair comparison between mcQQ-NET and QQ-NET, mcQQ-NET used the same training and testing scheme as QQ-NET [28] with one exception: noise consideration in data generation. While QQ-NET introduced Gaussian noise into the mGRE magnitude signals and QSM, mcQQ-NET incorporated Gaussian noise into the mGRE complex signals, offering a more realistic approach. In detail, mirroring QQ-NET, mcQQ-NET generated the training data using simulated stroke brains. First, the model parameters ($S_{0},R_{2},Y,v,\chi_{n}$) were estimated from real 34 stroke patient cases using QQ-CCTV [26] and were used as ground truth. The average (μ), standard deviation (σ), min, and max were $S_{0}$ (1.10, 0.04, 1.04, 2.12), R_2_ (19.6, 7.1, 7.3, 161.1 Hz), Y (0.67, 0.10, 0.31, 0.98), v (2.3, 1.2, 0.3, 7.2 %), and $\chi_{n}$ (−11.6, 37.5, −957.2, 159.7 ppb), respectively. $S_{0}$ was set to satisfy that the first echo magnitude signal was unity. Second, from the model parameter maps (ground truth), the mGRE complex signals were simulated for each brain voxel using Equations (1)–(3). Third, Gaussian noise was added to these complex signals to obtain SNR 20 (dB) at the first echo, with distinct noise instances for each training. This procedure produced pairs of ground truth (QQ-CCTV results) and simulated measurements (mGRE complex signals) for training. Out of 34 simulated datasets, 26, 2, and 6 were used for training, validation, and testing, respectively.

mcQQ-NET was implemented using Pytorch 1.13.0 [37] and NVIDIA RTX A6000 GPU. Minimization was performed using ADAM [38] with a learning rate of 10^−4^. Training was stopped at 400 epochs when the validation loss became stable. Due to GPU memory constraints, batch size was set to 1 with a 4D patch (16 × 200 × 200 × 48) as input, which approximately covers a whole brain. The patch center was randomly positioned within a selected brain, a process repeated for all training brains (1 epoch). Validation was carried out in a manner identical to the training process. For each epoch, the sequence of the training brains was randomly rearranged.

The trained mcQQ-NET was tested with the following three separate datasets, similar to the original QQ-NET [28]. Test Data 1: An additionally simulated stroke brain created using the same process as the training datasets (SNR 20 (dB)) (Figure 2). To reduce algorithm-dependent bias, the ground truth was set as the average of the QQ-NET and mcQQ-NET results from a real stroke patient (7 days post onset). This reconstruction was repeated five times with distinct instances of Gaussian noise to measure accuracy and precision. Test Data 2: A total of 30 ischemic stroke patients, a subset of the 34 patients devoid of hemorrhage and reperfusion, were divided into three groups based on the time interval between stroke onset and MRI scan [39]: acute (6–24 h, N = 4), subacute (1–5 days, N = 13), and chronic (≥5 days, N = 13) phases (Figure 3 and Figure 4). A five-fold cross-validation was performed to prevent overlap between training and test data [28]; six real patient brains were selected as test data, while the simulated datasets of the remaining 28 patients were utilized for training (N = 26) and validation (N = 2) data. This yielded 5 trained networks. The first trained network was used for Test Data 1 and 3. Test Data 3: Four healthy subjects scanned with identical imaging parameters (e.g., TE) to those used in training (Figure 5). The objective was to evaluate whether a network trained on simulated stroke brains could generate uniform OEF maps in healthy brains without introducing noticeable false positives, such as low OEF values typically observed in stroke lesions. During network testing, to ensure full brain coverage, patch sliding with 30% overlap was employed. Multiple overlapped patches were produced and subsequently combined to construct a single whole brain.

For real test data (Test Data 2 and 3), we compensated for the macroscopic field contribution in both magnitude and phase signal inputs. For the magnitude signal input at j’th echo ($\left|S_{j}^{\prime }\right|$), voxel spread function method [33] was used to estimate G: $|S_{j}^{\prime }|=\frac{\left|s_{j}\right|}{G_{j}}$, where $s_{j}$ is the mGRE complex. For phase signal input ($\phi_{j}^{\prime }$), the following steps were taken. First, the total ($f_{T}$) and background ($f_{B}$) fields were spatially unwrapped ($f_{T,uw}$ and $f_{B,uw}$) using a region-growing algorithm [40]. Second, unwrapped phase (${∠s}_{j,uw}$) was calculated using $f_{T,uw}$: ${∠s}_{j,uw}={∠s}_{j}-round\left(\frac{{∠s}_{j}-f_{T,uw}\cdot t_{j}-\phi_{0}}{2\pi }\right)\cdot 2\pi$, where $\phi_{0}$ is the initial phase at t=0. Lastly, the phase by tissue field ($\phi_{j}^{\prime }$) was obtained by compensating the contributions of background field ($f_{B,uw}$) and the initial phase ($\phi_{0}$): $\phi_{j}^{\prime }={∠s}_{j,uw}-f_{B.uw}\cdot t_{j}-\phi_{0}$.

We compared mcQQ-NET with QQ-NET [28], both of which were tested on the same test datasets. While they used identical training and testing scheme and data, they differed in in two aspects as mentioned earlier: network structure and model loss.

### 2.5. Statistical Analysis

For the simulation (Test Data 1), accuracy and precision were measured using mean error (ME) and mean standard deviation (MSD), respectively:(8)$$ME=\frac{1}{N_{v}}\sum_{i=1}^{N_{v}}OEF_{truth}-OEF_{avg}$$
and
(9)$$MSD=\frac{1}{N_{v}}\sum_{i=1}^{N_{v}}OEF_{std}$$
where $OEF_{avg}\equiv \frac{1}{N_{t}}\sum_{j=1}^{N_{t}}OEF_{i,j}$, $OEF_{std}\equiv \sqrt{\frac{1}{N_{t}}\sum_{j=1}^{N_{t}}{\left(OEF_{i,j}-\frac{1}{N_{t}}\sum_{j=1}^{N_{t}}OEF_{i,j}\right)}^{2}}$; i: the voxel index; j: the trial index; $N_{v}$: the number of voxels; $N_{t}$: the number of trials.

In the stroke patients (Test Data 2), lesion masks were constructed using DWI by an experienced neuroradiologist (S.J. with 7 years of experience). Corresponding contralateral normal tissue masks were created by mirroring the lesion mask to the other hemisphere and then trimmed by the same neuroradiologist. To measure lesion OEF abnormality, the OEF ratio was calculated between the lesion and its contralateral normal tissue ($OEF_{ratio}$) [41,42]:(10)$$OEF_{ratio}=\frac{{\bar{OEF_{lesion}}}_{\;}}{{\bar{OEF_{normal\;tissue}}}_{\;}}$$

$OEF_{ratio}$ was compared between QQ-NET and mcQQ-NET using a Wilcoxon signed rank test. A *p* value less than 0.05 was considered statistically significant.

In the healthy subjects (Test Data 3), structural similarity index (SSIM) was used to compare OEF maps from mcQQ-NET and QQ-NET quantitatively [43].

## 3. Results

Figure 2 shows the OEF comparison between QQ-NET and mcQQ-NET in the simulated brain (Test Data 1). mcQQ-NET provided higher accuracy, i.e., it was particularly better at capturing abnormally low OEF values within the lesion (smaller ME in the lesion: 1.13 vs. −0.34%).

**Figure 2 bioengineering-11-00131-f002:**
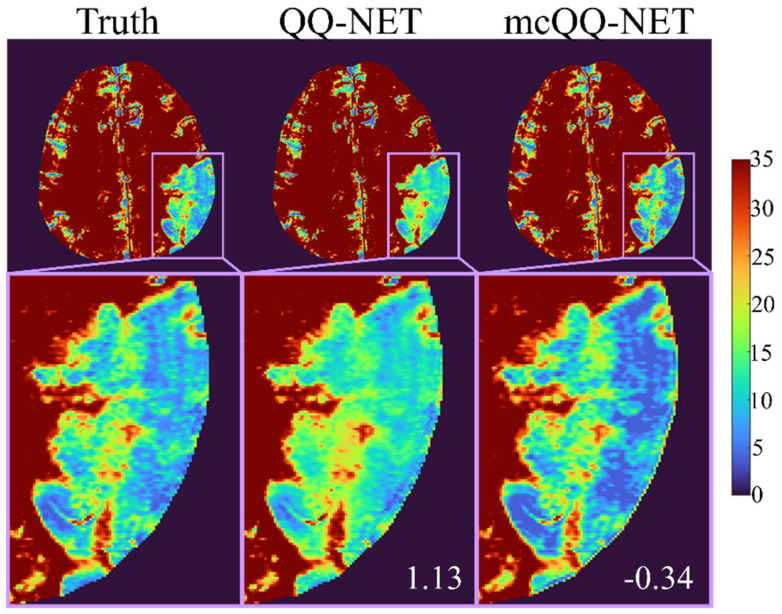
Comparison of OEF between QQ-NET and mcQQ-NET in a simulated brain (Test Data 1). The numbers in white indicate lesion mean error. The OEF maps are shown in the unit of %.

Figure 3 shows the ischemic stroke patients’ OEF maps (Test Data 2) generated by QQ-NET and mcQQ-NET. Compared to QQ-NET, mcQQ-NET showed lower lesion OEF values in the subacute phase and better spatial concurrence between low OEF regions and DWI-defined lesions in the chronic phase, e.g., 5 days post onset.

**Figure 3 bioengineering-11-00131-f003:**
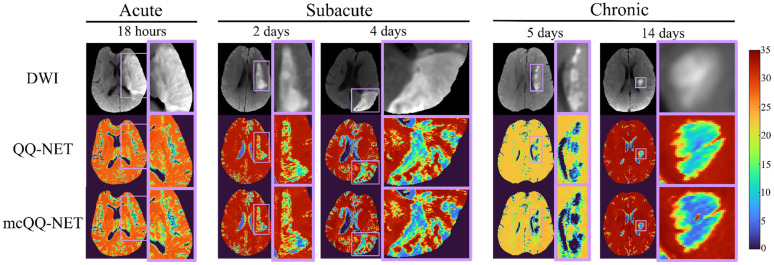
Comparison of OEFs estimated from QQ-NET and mcQQ-NET in 5 real ischemic stroke patients (Test Data 2). The OEF maps are shown in the unit of %.

Figure 4 presents box plots of the OEF ratio between the lesion and its contralateral normal tissue, $OEF_{ratio}$ (Test Data 2). mcQQ-NET provided significantly lower $OEF_{ratio}$ in the subacute phase compared to QQ-NET, 0.68 ± 0.24 vs. 0.66 ± 0.23 (*p* = 0.01). No significant difference was found in the acute (0.95 ± 0.13 vs. 0.97 ± 0.09, *p* = 0.63) and chronic (0.51 ± 0.20 vs. 0.54 ± 0.16, *p* = 0.34) phases.

**Figure 4 bioengineering-11-00131-f004:**
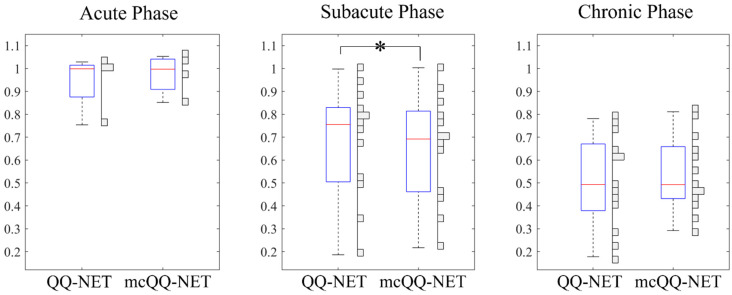
Box plots of OEF ratio between the lesion and its contralateral normal tissue in acute (6–24 h post onset, N = 4), subacute (1–5 days post onset, N = 13), and chronic (≥5 days post onset, N = 13) ischemic stroke patients. Red lines, blue boxes, and black whiskers indicate median, interquartile range, and the ra extending to 1.5 of the interquartile range, respectively. Asterisk (*) indicates a significant difference (*p* < 0.05, Wilcoxon signed rank test). Vertical histograms show the data distribution in each box plot.

In Figure 5, the OEF maps from QQ-NET and mcQQ-NET are shown for the four healthy subjects (Test Data 3). mcQQ-NET showed uniform OEF maps, similar to those of QQ-NET (SSIMs ≥ 0.99). However, QQ-NET exhibited a few extremely high OEF outliers near large veins (black arrows), which were not observed in mcQQ-NET.

**Figure 5 bioengineering-11-00131-f005:**
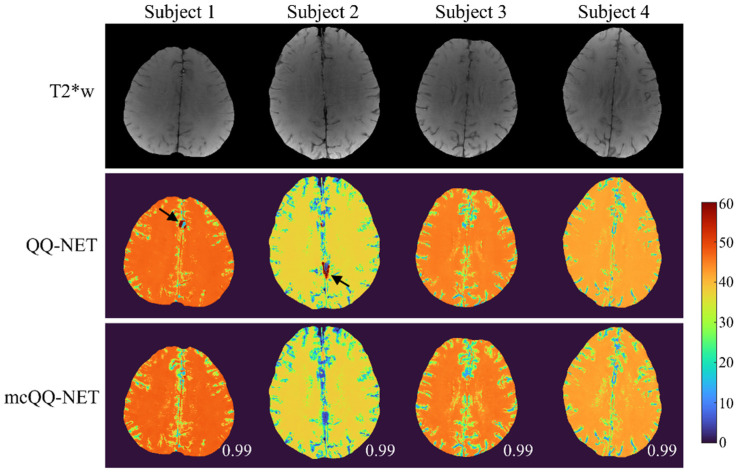
Comparison of OEF determined using QQ-NET and mcQQ-NET in four healthy subjects (Test Data 3). Numbers in white represent SSIM. The OEF maps are shown in the unit of %. Black arrows indicate extremely high OEF outliers located near large veins in QQ-NET.

## 4. Discussion

Our results demonstrate the feasibility of mcQQ-NET, a novel multi-echo complex signal modeling approach enhanced by deep learning. mcQQ-NET shows better OEF accuracy than QQ-NET [28], especially in low OEF lesions in simulations and in detecting low OEF abnormalities in subacute stroke patients. This improvement stems from realistically modeling Gaussian noise in mGRE complex signals and incorporating intermediate QSM processing steps into OEF quantification. mcQQ-NET also addresses clinical applicability issues in OEF mapping by using routinely applicable mGRE data, avoiding the complex and impractical procedures required by the current standard, ^15^O PET [13], and other MRI-based OEF techniques, including cfMRI [21,44] and T2-based methods [16,17].

Precise MRI signal modeling is essential for the accurate extraction of biophysics parameters, a principle well established in QSM [30] and now observed in OEF modeling in this study. In the simulations (Figure 2), mcQQ-NET shows better OEF accuracy, particularly in low OEF lesions. This improvement corresponds with an enhanced detection of OEF abnormalities in subacute stroke patients (Figure 3 and Figure 4). These results suggest that mcQQ-NET’s approach to modeling realistic noise can effectively tackle challenges such as decoupling OEF and *v*, a known issue in qBOLD-based techniques, including QQ [22]. The ability of mcQQ-NET to more accurately decouple these parameters leads to enhanced OEF estimation. This advancement in decoupling is further evidenced by mcQQ-NET’s improved accuracy in estimating *v* (Appendix A). Similarly, in real subacute stroke patients, mcQQ-NET’s reliable decoupling may result in low *v* values in ischemic lesions (e.g., 4 days post onset, Appendix A), which agrees with the expected blood volume decrease in ischemic stroke lesions [45].

In subacute stroke patients, mcQQ-NET consistently shows lower lesion OEF than QQ-NET (Figure 3). This is in line with the significantly lower OEF ratio between the lesion and its contralateral normal tissue, OEF_ratio_ (Figure 4), suggesting that mcQQ-NET may more accurately quantify OEF abnormalities. This is consistent with the expectation of low OEF_ratio_ in the subacute and chronic phases, as indicated in the PET literature [46]. The notably lower lesion OEF compared to the contralateral normal tissue might suggest irreversible damage in ischemic lesions [47]. In the acute phase, lesion OEF values are heterogeneous. For instance, in the case of 18 h post onset, mcQQ-NET shows OEF comparable to contralateral normal tissue for most of the DWI-defined lesion, but it indicates reduced OEF at the lower left corner of the lesion boundary (green/blue vs. red in Figure 3). This OEF heterogeneity might reflect rapid lesion progression within the first few hours following stroke onset [48]. Areas within the lesion with similar OEF to contralateral normal tissue may indicate recoverable tissue, while regions with lower OEF values could represent damaged tissues. The precise measurement of such OEF changes in the acute phase could be crucial for clinical decision making, such as evaluating the need for thrombosis treatment.

In the healthy subjects (Figure 5), mcQQ-NET shows uniform OEF maps, similar to QQ-NET, which agrees with previous PET studies [13,49]. These uniform OEF maps suggest that, despite being trained on simulated stroke brains with OEF abnormalities, both QQ-NET and mcQQ-NET can generate artifact-free OEF maps in healthy individuals. However, QQ-NET shows a few extremely high OEF outliers near large veins (black arrows). Given that the veins show low SNR due to rapid signal decay (veins have higher $R_{2}$ and lower $F_{BOLD}$ than brain tissue [19], leading to faster signal decay as per Equation (2)), these outliers might point to possible OEF estimation errors in low SNR areas. In contrast, mcQQ-NET does not show such high OEF outliers, suggesting that its more realistic noise modeling could enhance OEF accuracy in areas with low SNR, e.g., SNR ≤ 15 (dB), as calculated in our healthy subject dataset.

This study has some limitations that require further investigation. First, mcQQ currently employs a deep learning approach (mcQQ-NET) similar to QQ-NET [28]. Consequently, mcQQ-NET needs re-training for different TE sets, which poses a significant challenge for clinical applications. It may also be necessary to retrain mcQQ-NET for different imaging resolutions or SNRs, even though a previous QQ-NET study suggests insensitivity to these factors [28]. Second, a limited sample size could impact the training and testing of both QQ-NET and mcQQ-NET. They were evaluated on four acute, 13 subacute, and 13 chronic ischemic stroke patients, with notable differences observed only in the subacute phase. Testing on larger datasets may provide more comprehensive comparisons between QQ-NET and mcQQ-NET across all phases. For training, both QQ-NET and mcQQ-NET utilized identical training schemes and data, encompassing a wide physiological range for the model parameters, such as the entire possible range of OEF (0–100%). However, the limited number of 26 simulated stroke datasets might restrict the variety of parameter combinations. Training these models with a broader range of parameter combinations, including diverse physiological brain datasets, could yield more conclusive comparisons between QQ-NET and mcQQ-NET. Third, although an explanation for the improved OEF accuracy in mcQQ-NET compared to QQ-NET has been suggested—namely, the more realistic consideration of Gaussian noise in the complex mGRE signals using mcQQ—this explanation may not be sufficient, as it is not supported from rigorous analysis. Further investigation is required to establish a direct relationship between SNR and OEF accuracy. Finally, to determine whether mcQQ-NET offers improved OEF accuracy over QQ-NET in a clinically relevant context, mcQQ-NET should be tested against patients with OEF abnormalities, using the gold standard ^15^O-PET as a reference. Additionally, mcQQ-NET should also be compared with other well-investigated MRI-based OEF mapping techniques, such as cfMRI [14] or T2-based methods [16,17], to explore consistencies and discrepancies among these methods.

## 5. Conclusions

With its improved sensitivity to OEF abnormality based on more realistic biophysics modeling, mcQQ-NET holds potential for investigating tissue variability in neurologic disorders [50,51,52].

## Figures and Tables

**Figure 1 bioengineering-11-00131-f001:**
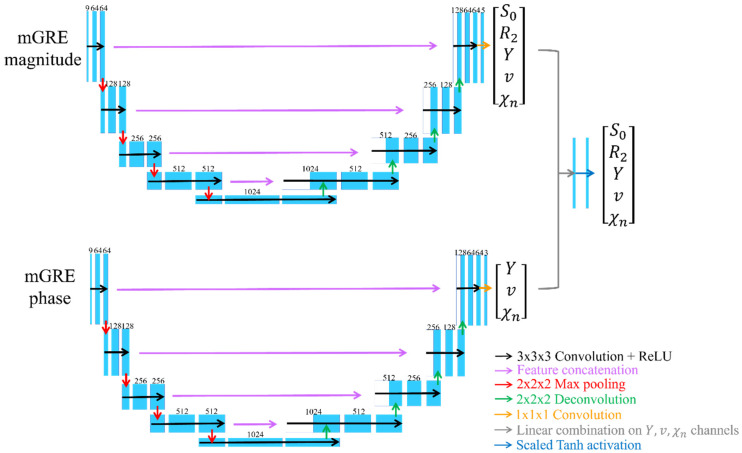
Network structure for mcQQ-NET. mcQQ-NET consists of two 4D Unets, one for mGRE magnitude and the other for mGRE phase input. Each Unet consists of an encoding and decoding path with 18 convolutional layers with 3 × 3 × 3 kernel (black), 4 max pooling layers with 2 × 2 × 2 kernel (red), 4 deconvolution layers with 2 × 2 × 2 kernel (green), 4 feature concatenations (purple), and 1 convolutional layer with 1 × 1 × 1 kernel (orange). Linear combination on the Y, *v*, χ_n_ channels of output of the two Unets (gray) and element-wise Tanh function were applied for setting parameter limit (blue).

## Data Availability

The datasets generated or analyzed during the study are available from the corresponding author upon reasonable request.

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
