# Peer review of "Multi-Echo Complex Quantitative Susceptibility Mapping and Quantitative Blood Oxygen Level-Dependent Magnitude (mcQSM + qBOLD or mcQQ) for Oxygen Extraction Fraction (OEF) Mapping"

_bioengineering, 2024, doi:10.3390/bioengineering11020131_

Round 1
Reviewer 1 Report
Comments and Suggestions for Authors
(1)Please do not cite large groups of papers without individually commenting on them. 35 references are cited the Para. 1, Section 1.
(2)Fig. 1 is unclear. Five parameters are output from magnitude model; while three parameters are output from phase model. How to combine them to generate the final five parameters?
(3)The authors should compare their method with more SOTA methods.
Comments on the Quality of English LanguageNone.
Reviewer 2 Report
Comments and Suggestions for Authors
This study introduces a novel multi-echo complex QQ approach (mcQQ) that accurately assumes Gaussian noise in the mGRE complex signals. The comments of this study are given as following.
1. The abstract focuses on describing the research background, methods and results. The abstract of this article must be rewritten.
2.The description of the Discussion section is too complicated and should be described in detail in a systematic and detailed manner.
3. Each symbol in eqs. (1) - (3) should be clearly defined, and the descriptions from Line 315 to 317 has to be expressed in formula form.
4. The text ",,,,,,,, provided uniform OEF maps, similar to QQ-NET, but ,,,,,,,, high OEF outliers in areas of low SNR. ,,,,,,, mcQQ-NET improves "OEF accuracy by better reflecting realistic data noise characteristics", what is the value of low SNR ? And what are the data noise characteristics?
5. It is recommended that the description from lines 149 to 158 and lines 219 to 229 be changed to a formula form.
6. The deep learning network and Figure 1 of this article must be systematically explained in detail.
7.The authors use the method provided by himself (mcQQ-NET) with the QQ-NET method to compare the OEF content. It is recommended to add other published literature to compare the pros and cons.
Reviewer 3 Report
Comments and Suggestions for Authors
General Comments
This paper presents a novel multi-echo complex QQ (mcQQ) that models realistic noise in mGRE complex signals. The proposed mcQQ was implemented using a deep learning framework (mcQQ-NET) and compared with the existing deep learning-based QQ (QQ-NET) in simulations, ischemic stroke patients, and healthy subjects. Both mcQQ-NET and QQ-NET used identical training and testing datasets and schemes for a fair comparison. In simulations, mcQQ-NET provided more accurate OEF than QQ-NET. This paper is quite detailed and technical, which is appropriate for a scientific context. However, here are some suggestions to enhance clarity and coherence:
Particular Comments:
1. Consider providing a brief introductory sentence outlining the importance of Oxygen Extraction Fraction (OEF) before diving into the technical details.
2. Clearly state the significance of accurate OEF estimation in the context of neurologic disorders.
3. It might be helpful to provide brief explanations or definitions for acronyms like QQ, QSM, and qBOLD upon their first mention to assist readers who may not be familiar with these terms.
4. Explicitly state the metrics used for comparing mcQQ-NET and QQ-NET in simulations, ischemic stroke patients, and healthy subjects.
5. Emphasize the potential impact on clinical decisions due to the improved accuracy provided by mcQQ-NET, reinforcing the practical implications of your study.
Comments on the Quality of English Language1. Flow and Structure:
Ensure a smooth transition between sentences and paragraphs. The text is technically dense, so maintaining a logical flow is crucial for reader comprehension.
2. Use of Language:
Simplify complex sentences where possible for easier comprehension, without sacrificing technical accuracy.
Round 2
Reviewer 2 Report
Comments and Suggestions for Authors
The more comments are suggested as following.
1. If w1 = 0.1 , w2 = 0.1, then w1 and w2 in the formula can be directly set to 0.1 values. Otherwise, the main functions of w1 and w2 should be explained. And E grad is not defined as well.
2. The authors do not provide a sufficient explanation that lower SNR values ​​lead to better research outcomes. At the same time, the lack of physical unit in the expression of SNR ≤ 30, is a serious scientific error.
3. The author’s responses to the reviewer’s comments are not shown in the article completly.
